# Automated Multi-Class Facial Syndrome Classification Using Transfer Learning Techniques

**DOI:** 10.3390/bioengineering11080827

**Published:** 2024-08-13

**Authors:** Fayroz F. Sherif, Nahed Tawfik, Doaa Mousa, Mohamed S. Abdallah, Young-Im Cho

**Affiliations:** 1Computers and Systems Department, Electronics Research Institute (ERI), Cairo 11843, Egypt; fayroz_farouk@eri.sci.eg (F.F.S.); nahedtawfik@eri.sci.eg (N.T.); doaa_mousa@eri.sci.eg (D.M.); 2Informatics Department, Electronics Research Institute (ERI), Cairo 11843, Egypt; 3AI Lab, DeltaX Co., Ltd., 5F, 590 Gyeongin-ro, Guro-gu, Seoul 08213, Republic of Korea; 4Department of Computer Engineering, Gachon University, Seongnam 13415, Republic of Korea

**Keywords:** rare diseases, facial recognition, genetic syndrome, artificial intelligence, deep learning

## Abstract

Genetic disorders affect over 6% of the global population and pose substantial obstacles to healthcare systems. Early identification of these rare facial genetic disorders is essential for managing related medical complexities and health issues. Many people consider the existing screening techniques inadequate, often leading to a diagnosis several years after birth. This study evaluated the efficacy of deep learning-based classifier models for accurately recognizing dysmorphic characteristics using facial photos. This study proposes a multi-class facial syndrome classification framework that encompasses a unique combination of diseases not previously examined together. The study focused on distinguishing between individuals with four specific genetic disorders (Down syndrome, Noonan syndrome, Turner syndrome, and Williams syndrome) and healthy controls. We investigated how well fine-tuning a few well-known convolutional neural network (CNN)-based pre-trained models—including VGG16, ResNet-50, ResNet152, and VGG-Face—worked for the multi-class facial syndrome classification task. We obtained the most encouraging results by adjusting the VGG-Face model. The proposed fine-tuned VGG-Face model not only demonstrated the best performance in this study, but it also performed better than other state-of-the-art pre-trained CNN models for the multi-class facial syndrome classification task. The fine-tuned model achieved both accuracy and an F1-Score of 90%, indicating significant progress in accurately detecting the specified genetic disorders.

## 1. Introduction

Genetic diseases and congenital anomalies affect approximately 2–5% of all live births, contributing to up to 30% of pediatric hospital admissions and accounting for almost 50% of childhood fatalities in developed nations [1].

A rare disease is characterized as a medical problem that impacts a small segment of the population. Around 4% of the world’s population is affected by these diseases. Although symptoms may not appear immediately, numerous uncommon diseases are genetically determined and endure throughout an individual’s lifespan. With ongoing improvements in genetic analysis, the ability to diagnose conditions is consistently getting better, which means that more testing is required. Nevertheless, it is important to be cautious when directing patients to clinical genetics centers due to the scarcity of resources, notably the shortage of skilled clinical geneticists. Significantly, between 30 and 40 percent of genetic illnesses are linked to dysmorphic traits, which are identifiable face abnormalities. A recent study has demonstrated the capacity of facial recognition technology to diagnose hereditary illnesses accurately.

According to the National Institutes of Health (NIH) [2], over 7000 uncommon genetic disorders have already been discovered worldwide. All of them are caused by a genetic mutation that may be inherited or acquired de novo and is the first occurrence in the family. Understanding the causes and characteristics of rare genetic syndromes aids in the diagnosis of individuals and families who are affected by a wide range of rare genetic syndromes and facilitates treatment methods [3].

Misdiagnosis is also a serious concern in pediatric genetics, just as it is in the diagnosis of rare diseases in individuals of all ages. But the faster children with rare diseases and their families relate to a diagnosis, the faster they can receive support for managing their condition. This improves outcomes for everyone.

Currently, accessing medical examinations remains challenging for individuals residing in several rural and impoverished regions due to the scarcity of medical resources. Consequently, this scarcity often results in delays in receiving necessary medical treatment. Even in large cities, there are still limits, such as the exorbitant cost, lengthy wait times at hospitals, and the inherent conflict between doctors and patients that sometimes results in medical disputes.

The list of medical software solutions that are changing the healthcare industry could go on. Some of the new opportunities that they are offering to healthcare professionals consist of task automatization, analyzing big data sets, reducing the workload, and enabling them to focus more on patients. Facial analysis technology is as good as qualified physicians in identifying syndromes. The use of facial analysis, deep learning, and artificial intelligence are making genetic analysis more accurate, more advanced, and more accessible, all at the same time.

Computer-aided detection of a genetic syndrome associated with a face phenotype is similar to facial recognition but presents additional obstacles, including data gathering difficulty and the delicate phenotypic patterns associated with many disorders. Doctors have been using technology as an aid, even though it is not meant to deliver a conclusive diagnosis. However, scholars assert that it raises various ethical and legal difficulties. These include ethnic bias in training datasets and commercial fragmentation of databases, which both have the potential to restrict the diagnostic tool’s reach. Earlier computer-aided syndrome detection methods have shown the potential to support doctors by analyzing photographs of patients’ faces. When combined with molecular analysis in clinical settings, these methods seem to complement next-generation sequencing (NGS) studies by inferring causal genetic variations from sequencing data. However, most of the research concentrates on differentiating healthy controls from affected people or identifying a few syndromes using limited pictures rather than tackling the real-world challenge of categorizing hundreds of syndromes from unconstrained images. Additionally, past research has employed small-scale training data, often up to 200 photos, which needs to be improved for deep-learning models. Due to the lack of a publicly available benchmark, it is not easy to evaluate the performance or accuracy of alternative approaches.

Recent advancements in computer vision have led to the development of many apps that utilize artificial intelligence to aid in the identification of uncommon genetic illnesses. These programs analyze two-dimensional frontal photographs of patients to assist the diagnostic process. Several of these features have been integrated into Internet platforms with easy-to-use interfaces and facial analysis services, such as Face2Gene [4].

Unfortunately, users cannot perform face analysis operations internally due to the unavailability of both the training data and the trained models. The available datasets used in facial diagnosis research are private and usually very small compared to datasets used in other machine learning applications. Deep transfer learning involves utilizing the knowledge and learned representations obtained from solving one problem and applying them to a different but related problem [5]. By leveraging pre-trained models that have been trained on large-scale datasets for general facial recognition tasks, we can benefit from the rich feature representations and generalization capabilities already acquired. This approach allows us to mitigate the issue of insufficient training data for facial diagnosis by leveraging knowledge gained from the broader domain of facial recognition.

By incorporating deep transfer learning into our research, we aim to take advantage of the prior knowledge and expertise captured in pre-trained models to enhance the performance and robustness of our classification systems for syndrome identification. This approach enables us to benefit from the wealth of information learned from large-scale datasets. It empowers our models to generalize well, even in scenarios where the training data for specific syndromes is limited. These models provide feature vectors with many dimensions, which allows for the detection of both known and unknown illnesses.

In this paper, we focus on designing an approach in the field of syndrome identification based on facial images. Unlike previous studies that primarily employ binary classifiers, our approach is designed as a multi-classifier system. We aim to classify facial images into specific syndrome categories, including Down syndrome, Noonan syndrome, Williams syndrome, Turner syndrome, or normal cases. To the best of our knowledge, we are the first to introduce these specific combinations of syndrome diseases in a multi-classification scenario. To enhance the performance of our classification system, we intend to leverage the capabilities of pre-trained models such as ResNet-50, VGG16, ResNet-152, and VGG-Face to mitigate the issue of insufficient training data for facial diagnosis. By adopting a unified approach that considers all syndromes together in one classifier, we anticipate significant improvements in both the efficiency and accuracy of our system.

The main contributions of this study are as follows:The proposed multi-class facial syndrome classification framework encompasses a combination of diseases that have not been studied together previously, to the best of our knowledge. This unique approach extends the current body of knowledge in the field of automated facial syndrome diagnosis.In this study, we have thoroughly investigated the efficacy of fine-tuning several popular convolutional neural network (CNN)-based pre-trained models, including VGG16, ResNet-50, ResNet152, and VGG-Face, for the task of multi-class facial syndrome classification. This comprehensive evaluation provides valuable insights into the performance of transfer learning techniques for this specialized medical application.The proposed fine-tuned VGG-Face model demonstrated superior performance compared to other state-of-the-art pre-trained CNN models for the multi-class facial syndrome classification task. This finding highlights the potential of leveraging domain-specific pre-trained models for improved diagnostic accuracy.To address the inherent challenge of class imbalance in the facial syndrome classification dataset, we implemented an effective data augmentation technique. This approach enabled the artificial generation of additional training samples for each syndrome class, thereby enhancing the model’s ability to learn discriminative features and generalize to unseen data.This study validates the applicability of deep transfer learning methods for the classification of multiple facial syndromes, even when dealing with a relatively small dataset. The successful deployment of these techniques in this medical domain contributes to the growing body of evidence supporting the utility of transfer learning in healthcare applications.

The rest of this paper is organized as follows: Section 2 reviews the related work of computer-aided facial diagnosis. Section 3 describes our proposed methodology. Section 4 analyzes and discusses the experimental results. Section 5 addresses a conclusion.

## 2. Related Work

Several studies have been carried out to analyze and identify different genetic syndromes based on facial features. Machine learning approaches have been extensively studied for rapidly diagnosing a wide range of genetic disorders. Zhao et al. [6] introduced a machine learning-based approach for the automatic detection of Down syndrome. Local binary patterns were used to extract geometric and texture information in the vicinity of the landmarks. The highest level of performance was attained with an accuracy of 95% by utilizing a support vector machine (SVM) with a radial basis function kernel. The study conducted by Cerrolaza et al. [7] introduced a comprehensive framework for identifying various genetic abnormalities, such as Down syndrome, using a database of 145 facial images. This framework combined both geometric and textural characteristics. This study suggested the integration of morphological and local appearance variables by employing local binary patterns, resulting in a detection accuracy of 94%. Dima et al. [8] employed facial recognition techniques to extract features and utilized SVMs and K-nearest neighbors classifiers to identify the existence of Down syndrome. The researchers utilized 50 samples of individuals with Down syndrome and obtained accuracy findings ranging from 93% to 100% by employing various datasets for the control group. Kumov et al. [9] investigated the utilization of 3D face reconstruction, in addition to geometric and deep features, to classify eight genetic disorders. The researchers utilized a database consisting of 1462 samples in total and attained an average accuracy of 92%. Pantel et al. [10] aim to assess DeepGestalt’s accuracy with images of people with and without a genetic syndrome using linear SVM. Images were collected from online reports and were obtained directly by the authors for this study. The classification experiments were implemented based on all photos and images of white persons and persons of other ethnicities to evaluate the effect of ethnicity on SVM performance. The SVMs automatically distinguish between the two classes of images by relying on DeepGestalt’s result lists. The results prove that DeepGestalt has high sensitivity (top 10 sensitivity: 295/323, 91%).

Recently, deep learning approaches have demonstrated excellent performance in benchmark face identification datasets. The technology is employed in numerous medical image analysis applications, yielding excellent outcomes and potentially serving as a highly accurate diagnostic approach for genetic syndromes. Qin et al. [11] proposed a Down syndrome identification system based on unconstrained two-dimensional face images using CNN. The training phase of CNN was implemented in two main steps. Firstly, a general facial recognition network was formed utilizing a large-scale face identity database. Then, a dataset of Down syndrome and healthy control (HC) images selected from public databases were used as trained (70%) and tested (30%). Their CNN-based algorithm achieved 95.87% accuracy, 93.18% recall, and 97.40% specificity in Down syndrome classification. Authors in [12] implemented a computer-aided facial diagnosis of different diseases using deep transfer learning. The proposed method was implemented using two different approaches: a binary classification (beta-thalassemia vs. normal) and multi-classification (Down syndrome, beta-thalassemia, hyperthyroidism, leprosy, and normal) with a small dataset. The dataset used includes disease-specific face images, which were collected from medical forums, professional medical publications, medical websites, and hospitals with definite diagnostic results. The proposed method’s accuracy reaches over 90%, which outperforms the performance of both traditional machine learning methods and clinicians in the experiments. Gurovich et al. [13] proposed a facial analysis framework for genetic syndrome classification called DeepGestalt, which determines similarities between hundreds of syndromes. This method applies deep learning and learns facial representation from a large facial dataset, and then knowledge transfer to the genetic syndrome domain is implemented through fine-tuning.

The final experiment shows that DeepGestalt achieved 91% top 10 accuracy in determining the correct syndrome in 502 different images. The study in [14] proposed an automated facial identification system that trained on images of 127 Noonan syndrome (NS) patients, 163 healthy children, and 130 children with other dysmorphic syndromes. Thirty-seven NS patients were recruited from Guangdong Provincial People’s Hospital. A CNN framework with an additive angular margin (ArcFace) loss function was built. Two traditional machine learning methods and a CNN with a cross-entropy loss function were also implemented. Transfer learning and data augmentation were utilized in the training phase. The identification performance was evaluated by five-fold cross-validation. A comparison of the proposed method, the two traditional machine learning methods, and six clinicians was achieved.

Results: When distinguishing NS patients from healthy individuals, the CNN-Arcface model accomplished an accuracy of 92%. Furthermore, a VGG16 screening scheme for the identification of genetic syndromes was proposed [15]. A group of 456 frontal facial images was curated from 228 children with genetic syndromes and 228 photos of healthy children in Guangdong Provincial People’s Hospital. The performance of the VGG-16 scheme was assessed by five-fold cross-validation. Also, a comparison of the VGG-16 scheme with five clinicians was executed. The proposed method achieved the highest accuracy of 88.60%, specificity of 91.24%, and recall of 85.97%. Mishima et al. [16] evaluated the effectiveness of the AI-powered facial recognition system, Face2Gene, in diagnosing congenital dysmorphic syndromes, such as Down syndrome, in a group of Japanese patients. Among the 74 patients in group one, who had a total of 47 different congenital dysmorphic disorders, facial recognition failed in four instances; out of the 70 patients who were successfully recognized by face recognition, between 13 and 21 had disorders that were not part of the training set of Face2Gene. Out of the remaining 49 individuals, the accurate syndrome was among the top 10 recommended diagnoses in 85.7% of cases. Within group two, which consisted of 34 individuals diagnosed with Down syndrome, the algorithm successfully identified Down syndrome as the most prominent condition for the face photos of the youngest subjects, spanning from newborns to 25-year-olds. From facial photos of patients of 20 years or older, Down syndrome was identified as the most prevalent or second most prevalent disorder in 82.2% and 100% of cases, respectively. The results suggest that the present iteration of Face2Gene is a good instrument for aiding clinical geneticists in prioritizing potential syndromes in Japanese patients. Nevertheless, the study proposes that augmenting the training dataset with a broader range of ethnic origins could improve the system’s performance. 

Liu et al. [17] presented an automated facial recognition model for detecting Williams–Beuren syndrome (WBS) by utilizing CNNs. They obtained frontal facial photographs from a total of 104 children diagnosed with WBS, 91 children diagnosed with different genetic abnormalities, and 145 healthy controls. The authors employed transfer learning from ImageNet to create five facial recognition frameworks using various CNN architectures: VGG16, VGG19, ResNet-18, ResNet-34, and MobileNet-V2. The VGG19 architecture displayed the best performance with an accuracy of 92.7%, whereas MobileNet-V2 obtained the lowest accuracy with an accuracy of 85.6%. Attallah [18] suggested a computer-aided facial diagnostic method, FaceDisNet. Several spatial deep features from CNNs with different architectures are integrated to create FaceDisNet. It extracts spatial-spectral characteristics in addition to relying solely on spatial features. To minimize the enormous dimension of features arising from feature fusion, it uses two feature selection strategies. Ultimately, it performs classification by constructing an ensemble classifier using DenseNet-201, Inception V3, ResNet-50, and ResNet-101. FaceDisNet’s performance attests to identifying both single and multiple disorders. Following the binary and multiclass classification categories’ ensemble classification and feature selection stages, FaceDisNet’s highest accuracy was 98.57% and 98%, respectively. Authors in [19] utilized a CNN based on ResNet architecture to train a model for diagnosing Turner syndrome (TS). They used a dataset consisting of 170 pictures of TS patients and 1053 photographs of controls. This model achieved an average accuracy of 96%. The authors also examined the effect on the classification performance when training models using age-matched or non-age-matched samples, but no statistically significant difference was seen. The same phenomenon occurred for both samples that were matched in height and those that were not matched in height.

Additionally, authors in [20] presented an automated classification system for TS using unsupervised feature learning. This method for diagnosing TS depended on manually designed picture features and support vector machines. The result achieved an accuracy of 84.95%. Yang et al. [14] introduced a CNN that utilizes the ArcFace loss function. This model detected children with Noonan syndrome. Following training with a Casia-WebFaces dataset, the network underwent fine-tuning with pictures of Noonan syndrome, various dysmorphic syndromes, and controls. The obtained accuracy was 92%. A method that utilizes transfer learning techniques to analyze face images and detect Down syndrome in children at an early stage was addressed in [21]. The approach combines the VGG16, non-negative matrix factorization (NMF), and light gradient boosting machine (LGBM) techniques. Firstly, spatial attributes are obtained from a dataset of facial images using the VGG16 model. Afterward, a combined set of features is created using non-negative matrix factorization (NMF) and a light gradient boosting machine (LGBM) for the extracted spatial features. The algorithm achieved an accuracy of 99% using a dataset consisting of 1500 Down syndrome patients and 1509 normal controls.

Deep facial diagnosis and the integration of all biomedical knowledge are expected to aid clinical geneticists in their decision-making. However, most of the research concentrates on differentiating unaffected from affected people (binary classification) or identifying a few syndromes using limited pictures rather than tackling the real-world challenge of categorizing hundreds of syndromes from unconstrained images. Additionally, the majority of these algorithms rely on computational molecular biology and manually developed feature extraction methods. Nevertheless, deep learning methods are advantageous because they can classify images without the need for image processing or feature extraction procedures. Few of the earlier studies used deep learning approaches alone for categorization.

Furthermore, tiny and private datasets have been used in many previous studies. It is impossible to assess the effectiveness of such earlier methods because there is not a publicly available benchmark dataset online. Table 1 summarizes some studies that have addressed the diagnosis of facial disease.

## 3. Materials and Methods

### 3.1. Dataset

We obtained our dataset from two primary sources: the GestaltMatcher database (GMDB) and publicly accessible platforms on the Internet. For the genetic disorder samples, we utilized the GestaltMatcher database (GMDB), which was built by Hsieh et al. [22,23] to collect medical images of rare disorders from publications and patients with proper consent from clinics. The GMDB is an open resource for clinicians and researchers working in the medical research field, although applicants need to be reviewed by a committee before accessing the database. The GestaltMatcher database is a service provided by the Association for Genome Diagnostics (AGD), a recognized non-profit organization based in Germany. The goal of GMDB is to enhance the transparency and availability of scientific discoveries and promote collaboration among academics and practitioners.

We initially generated a database of 1146 photographs sourced from GMDB and publicly accessible platforms on the Internet. This database included 485 images of healthy controls (HC), as well as 661 images of individuals with four specifically recognized developmental diseases, as reported in Table 2. Systematic examinations were conducted to exclude photographs in which the face or an eye was not distinctly apparent or in which an expert clinician could not confirm the diagnosis of the genetic disorders or the absence of any disorders. In addition, the distribution of the dataset in terms of age (mean 6 years, SD 4.58 years; range 0.6–13 years), gender, and ethnicity is displayed in Figure 1.

### 3.2. Syndrome Identification Pipeline

The diagram in Figure 2 represents the proposed multi-class syndrome classification algorithm, which contains the main steps such as image preprocessing, facial detection, image augmentation, transfer learning, syndrome classification, and model evaluation.

#### 3.2.1. Image Preprocessing

The experimental datasets exhibited significant heterogeneity in terms of image dimensionality and categorical representations. A common preprocessing technique applied across these diverse inputs was to standardize the visual input by resizing all images to a fixed spatial resolution compatible with the architectural requirements of the respective CNN models under evaluation. For the facial localization stage, we employed the Haar cascade classifier face_cascade implementation from the OpenCV computer vision library [24]. This step identified and localized the faces within the images. We then proceeded to crop the face region from the original image based on the coordinates of the bounding box. By doing so, we can concentrate entirely on the subject’s face and avoid any background noise or distractions. Finally, picture resizing was carried out to make sure everything was compatible and uniform throughout the dataset. To avoid distortion, the images were resized to a consistent resolution while preserving the aspect ratio. This step is essential to ensure that the input dimensions used to train and test the classification models are consistent. The resulting cropped and resized face images can then be further processed or used as input for subsequent analysis and classification tasks. Python libraries like TensorFlow and Keras are used to develop and simulate these proposed methodologies.

#### 3.2.2. Data Augmentation

Data augmentation is a widely adopted technique used to artificially expand the size and diversity of training datasets, particularly in the context of computer vision applications [25]. The generation of additional training samples through the application of various transformations is crucial for establishing the efficacy of deep learning-based models, which typically require extensive amounts of diverse data to achieve robust predictive performance. In this study, we employed a two-pronged data augmentation strategy encompassing both spatial and color-based transformations [26]. The spatial augmentation procedures included scaling, cropping, shearing, padding, flipping, translation, and rotation. The color enhancement techniques involved adjusting brightness, saturation, and contrast. Expressly, the shear range, zoom range, height shift range, and width shift range were all set to 0.2, and the rotation range was set to 40 degrees. Additionally, a horizontal flip was applied using the ‘nearest’ fill mode.

The color enhancement techniques involved adjusting brightness, saturation, and contrast. The brightness range was set from 0.5 to 1.5, allowing for both darkening and brightening of the input images.

These augmentation operations were selectively applied to the input images based on the target object categories. Specifically, we applied more aggressive transformation parameters to the ‘Down’ and ‘Turner’ classes, generating 15 augmented samples per original image. In contrast, the ‘Williams’ and ‘Noonan’ categories received a more moderate level of augmentation, with five additional samples created per input. For the ‘healthy controls’ class, we only generated two augmented versions per original image. This selective augmentation strategy was designed to address the inherent class imbalance present in the dataset, with more aggressive transformation parameters utilized for the minority classes (‘Down’ and ‘Turner’). Conversely, the majority classes ‘Williams’ and ‘Noonan’ classes, as well as the ‘healthy controls’ category, received a more conservative degree of augmentation to address the class imbalance and improve model generalization on those less prevalent samples.

#### 3.2.3. Proposed Deep Learning Framework

We recognize that training a CNN architecture from scratch may result in overfitting due to the limited availability of training data for the task of facial syndrome diagnosis. To overcome this challenge, we propose to leverage the powerful technique of deep transfer learning.

Deep transfer learning involves utilizing the knowledge and learned representations obtained from solving one problem and applying them to a different but related problem [27]. By leveraging pre-trained models trained on large-scale datasets for general facial recognition tasks, we can benefit from the rich feature representations and generalization capabilities already acquired. There are two main techniques to implement transfer learning: either by training the model from the top layers or by freezing the top layers and then fine-tuning it on the new dataset.

Because the proposed model considers five distinct classes, it employs the second method, which involves freezing the model from the top layers and then fine-tuning it on our facial images. Pretrained facial syndromes classification models like VGG16, ResNet-50, ResNet-152, and VGG-FACE can overcome insufficient training data. These pre-trained facial recognition models can help transfer knowledge to syndrome classification.

The VGG16 model, with its simple and uniform architecture of small 3 × 3 filters and max pooling layers, can capture fine-grained facial details, making it a promising syndrome classification candidate [28]. However, VGG16’s many parameters can increase computational and memory needs during training and inference.

ResNet-50, with its innovative residual connections, performs well on image classification tasks, including complex facial features. Residual learning solves the vanishing gradient problem, allowing more profound network training [29]. ResNet-50 may require more computational resources than shallower models, but its feature extraction may benefit multi-class syndrome classification.

ResNet152 deepens ResNet’s architecture to enhance its ability to capture intricate facial patterns and features. Model complexity can improve classification accuracy, especially with large and diverse datasets [30,31]. However, the ResNet152 model’s computational demands may make training and deployment difficult, especially in resource-constrained environments.

The VGG-FACE model, pre-trained on 982,803 online photos of 2622 famous people, may be helpful for facial syndrome classification. The architecture of VGG-Face includes thirteen convolutional layers and three fully linked layers (source: https://www.robots.ox.ac.uk/~vgg/software/vgg_face/ accessed on 1 May 2024). By fine-tuning the VGG-FACE architecture to the specific requirements of the syndrome diagnosis task, we can leverage the domain-specific knowledge embedded in the pre-trained model, potentially outperforming the performance of more generic pre-trained models.

### 3.3. The Model Architecture and Training Strategy

Following the base pre-trained model, a global average pooling layer is added to the architecture. This layer reduces the spatial dimensions of the output from the base model to a fixed size, regardless of the input image size. This helps in reducing the number of parameters and computational complexity, making the model more efficient. After the global average pooling layer, three dense layers with sizes 1024, 512, and 256 are added. The ReLU activation function is used for these dense layers, which introduces non-linearity to the model and helps learn complex relationships between facial features and syndrome classes.

To avoid overfitting, a dropout layer is included after the dense layers. This layer has a dropout rate of 0.2 and is used to randomly remove some of the network’s connections during training, regularizing the model and improving its generalization performance. The final layer of the model is a highly dense layer employing a SoftMax activation function. This layer has five units corresponding to the five classes in the dataset: ‘Down’, ‘Noonan’, ‘Turner’, ‘Williams’, and ‘healthy controls’. The SoftMax activation function ensures that the model output represents syndrome class probability distributions. All the pre-trained model architectures used in the research—VGG16, ResNet-50, ResNet152, and VGG-FACE—used identical setups and methods.

Regarding the dataset division and training parameters, the dataset used in all models is related to a syndrome project involving images categorized into the five classes mentioned earlier. To reduce overfitting, the dataset is split into 70% training, 10% validation, and 20% testing sets. This provides enough data for model training, validation, and performance evaluation of new data.

The models are trained using categorical cross-entropy loss, a commonly used criterion in multi-class classification tasks. This loss function measures the discrepancy between the predicted class probabilities and the actual class labels. The selected optimizer is Adam, a popular optimizer that adjusts learning rates during training and captures the benefits of AdaGrad and RMSProp [24]. These training parameters optimize model performance during training, ensuring efficient learning and accurate classification.

#### Optimizing Hyperparameters

Specifying hyperparameters is crucial before training a model. They influence model behavior and performance. Hyperparameters encompass several factors that influence a model’s performance, such as the dropout rate, activation function, number of neurons in hidden layers, number of epochs, and batch size [21].

For optimization purposes, the hyperparameters were tuned through a process of trial and error. An adaptive learning rate approach was implemented by incorporating an exponential decay learning rate scheduler. This scheduler decreases the learning rate exponentially throughout training. Early stopping was also employed to prevent overfitting during the training process. Early stopping is a technique that monitors the model’s performance on a validation set during training and stops the training process when the validation performance stops improving. Table 3 reports the specific values selected for each hyperparameter, ensuring transparency and replication of the experimental setup.

### 3.4. Performance Evaluation

To evaluate the performance of the different implemented models, the test dataset is utilized to assess the model’s accuracy and effectiveness in classifying images. In this process, raw frontal facial images, which were not part of the training dataset, underwent preprocessing and were input into the multi-syndrome identification network. The output represented the similarity between the input image and each syndrome or healthy face within the training dataset, thereby indicating the likelihood of the individual having each syndrome. The model’s predictions are compared with the actual labels from the test dataset to evaluate its accuracy and performance. The models’ performance is assessed by calculating metrics like accuracy and F1 score and generating a confusion matrix to visualize classification results.

**Accuracy** measures the proportion of correctly classified images out of the total number of images in the test dataset [32].

**Precision** is a measure of how accurately a model can identify positive samples among the predicted positives. The calculation determines the proportion of correct optimistic predictions out of the total number of optimistic predictions, including both correct and incorrect ones. Precision emphasizes the dependability of affirmative outcomes. Predictions can be represented by the formula Precision = TP/(TP + FP), where TP represents the accurate positive predictions, and FP represents the false optimistic predictions.

**Recall**, also referred to as sensitivity or the true positive rate, quantifies the model’s capacity to accurately detect positive samples from all the actual positives. The calculation determines the proportion of correct positive predictions to the total number of positive predictions, including both correct and incorrect ones. The concept of recall centers around the degree to which positive predictions are accurately identified. The equation for it is R = TP/(TP + FN).

**F1 Score** is a metric that combines precision and recall, providing a balance between them. It is particularly useful when dealing with imbalanced datasets [33]. The F1 score is calculated as follows F1 = (2 × P × R)/(P + R) and scaled from 0 to 1, where a score of 1 indicates the best performance.

**Support value** indicates the number of instances or samples for each class in the dataset used to train and evaluate the model.

**Confusion Matrix** is the table that summarizes the model’s performance by comparing actual class labels with predicted class labels. It provides insights into true positives, true negatives, false positives, and false negatives [33].

**Loss** calculates how well the model is performing during training and validation. A lower loss indicates better model performance [34].

## 4. Results and Discussion

The goal of the study was to distinguish between healthy controls and those who had one of four specific genetic disorders: Down syndrome, Noonan syndrome, Turner syndrome, or Williams syndrome. We examined the effectiveness of fine-tuning a few popular pre-trained models based on convolutional neural networks (CNNs), such as VGG16, ResNet-50, ResNet-152, and VGG-Face, for the multi-class facial syndrome classification problem. Data augmentation has been used to balance the dataset for the training phase, as shown in Figure 3. The augmentation process resulted in the generation of 4861 image samples, yielding a balanced dataset of 5776 images in total. A set of 231 distinct images were reserved for the final testing phase. By expanding the training data through these carefully designed augmentation strategies, we aimed to expose the deep learning models to a wider range of visual variations, thereby enhancing their ability to generalize to unseen samples and handle diverse real-world scenarios.

### Model Performance

Our investigation into this multi-classifier approach, utilizing different pre-trained models, contributes to the field of syndrome identification from facial images. In our research study, we compared the performance of four pre-trained models, namely, VGG16, ResNet-50, ResNet-152, and VGG-Face, for facial syndrome classification. Table 4, Table 5, Table 6 and Table 7 present the results of the classification task using the different pre-trained models. The tables show the test loss, test accuracy, and F1 score for each model. The results indicate that all four pre-trained models achieved reasonably good performance in classifying the syndromes. However, there are some notable differences in their performance metrics.

On the other hand, VGG16 exhibited slightly lower performance compared to the other models. It had a higher test loss, indicating less accurate predictions, as well as a lower test accuracy and F1 score, indicating lower overall classification performance. Comparing ResNet-50 and ResNet-152, both models performed well, but ResNet152 demonstrated superior performance across all metrics. Compared to ResNet-50, it achieved a lower test loss, higher test accuracy, and a higher F1 score. These results suggest that the depth and complexity of the pre-trained models play a significant role in their classification performance. Models with more layers and parameters, such as ResNet152, have the potential to capture more intricate patterns and features, leading to improved classification accuracy. ResNet-50 also achieved reasonably good results and is a more suitable option in scenarios with limited computational capabilities. 

The proposed fine-tuned VGG-Face model not only demonstrated the best performance in this study, but it also performed better than other state-of-the-art pre-trained CNN models for the multi-class facial syndrome classification task. The VGGFace model’s superior performance could be attributed to its training on large face datasets with millions of face images, which allowed the model to acquire universal facial representations that performed well on new data. Overall, the results highlight the effectiveness of pre-trained models in syndrome classification and emphasize the importance of selecting an appropriate model architecture to achieve optimal performance. Figure 4, Figure 5, Figure 6 and Figure 7 illustrate the performance of the VGG16, ResNet-50, ResNet-152, and NGG-Face models, respectively. The VGG16 model performance in Figure 4, while having strong training performance, exhibits lower performance than the other models on the validation and test sets. This suggests that the VGG16 model is more prone to overfitting compared to the other, more complex models.

The performance metrics for the different CNN algorithms on the various classes are provided in detail as follows:**Down syndrome:**○The VGG16 model had the highest precision, indicating it was good at correctly identifying Down syndrome cases.○However, its low recall suggests it missed a significant number of Down syndrome cases.○The ResNet-50 model had a more balanced performance, with moderate precision and higher recall. This means it was able to identify a more significant proportion of Down syndrome cases, though with more false positives than VGG16.○The ResNet-152 model’s performance was between the other two ResNet models, indicating a reasonable compromise between precision and recall.○The VGG-Face model had the highest recall, suggesting it was the most effective at detecting Down syndrome cases, even if it had a slightly lower precision than ResNet-152.**Turner syndrome:**○The VGG16 model had the lowest precision but the highest recall, indicating it was good at identifying Turner syndrome cases but also had a higher rate of false positives.○The ResNet-50 model had a more balanced performance, with higher precision and recall than VGG16.○The ResNet-152 model had the best precision, suggesting it was the most accurate in identifying Turner syndrome cases, but its recall was lower than the other models.○The VGG-Face model performed better than the other models, with a good balance between precision and recall.**Williams syndrome:**○The VGG16 model had a good balance between precision and recall, indicating it was effective at identifying Williams syndrome cases.○The ResNet-50 model had the highest precision, meaning it was the most accurate in identifying Williams syndrome cases, but its recall was slightly lower than VGG16.○The ResNet-152 model had the best overall performance, with high precision and recall, resulting in the highest F1 score for this class.○The VGG-Face model had a good performance but was slightly lower than the ResNet models in terms of both precision and recall.**Noonan syndrome:**○The VGG16 model had a reasonable performance, with a good balance between precision and recall.○The ResNet-50 model had the highest precision, suggesting it was the most accurate in identifying Noonan syndrome cases.○The ResNet152 model had the highest recall, indicating it was the most effective at detecting Noonan syndrome cases, though its precision was slightly lower than ResNet-50.○The VGG-Face model had the best overall performance, with high precision and the highest recall, resulting in the best F1 score for this class.**Healthy controls class:**○All the models performed very well in the healthy controls class, with high precision and recall.○The ResNet-50 model performed best, with near-perfect precision and recall, resulting in the highest F1 score.○The ResNet-152 and VGG-Face models also had excellent performance, with very high precision and recall.○The VGG16 model performed well but was slightly lower than the other models in terms of both precision and recall.

In addition, for the purpose of comparing the deep learning models, we created five comparative ROC-AUC charts (Figure 8, Figure 9, Figure 10, Figure 11 and Figure 12), one for each class. Overall, the analysis suggests that the VGG-Face model generally had the best performance across the different syndrome classes, with a good balance between precision and recall. The ResNet models, particularly ResNet-50 and ResNet-152, showed more robust performance in the healthy controls class. The choice of the best model may depend on the specific needs and requirements of the application, such as the importance of precision vs. recall for each syndrome class.

The experiments were conducted using Python on a PC with an Intel Core i9-12900H CPU and DDR4 16GB of RAM, running Windows 11 Pro 64-bit. The execution time of the proposed CNN models was computed and compared to one another. Table 8 displays the operational efficiency of CNN models, specifically the average durations for training and inference. The VGG-Face model exhibited the shortest inference time.

Most previous research has focused on either differentiating normal from affected individuals or identifying a few specific syndromes using limited image datasets. In contrast, this study tackles the more challenging real-world problem of classifying multiple genetic syndromes (Down syndrome, Noonan syndrome, Turner syndrome, and Williams syndrome) simultaneously using a multi-class classification approach—a unique combination of diseases not previously examined together, to the best of our knowledge. However, the comparative analysis of the proposed framework with other studies remains complex due to the inherent differences in the datasets and classification tasks across the published literature. The datasets used can vary significantly, and the number of syndrome classes considered also tends to differ considerably. This lack of a standardized benchmark dataset and the diversity in the number of syndrome classes studied pose a significant challenge for the research community. Addressing these issues would enable meaningful comparisons between the various facial syndrome classification frameworks reported in the literature.

## 5. Conclusions

This study introduces the concept of deep transfer learning, which involves applying face recognition techniques to achieve accurate computer-assisted facial diagnosis. The study proposes a multi-class facial syndrome classification framework that encompasses a unique combination of diseases not previously examined together. The study focused on distinguishing between individuals with four specific genetic disorders (Down syndrome, Noonan syndrome, Turner syndrome, and Williams’s syndrome) and normal. We investigated how well fine-tuning a well-known convolutional neural network (CNN)-based pre-trained models—including VGG16, ResNet-50, ResNet-152, and VGG-Face—worked for the multiclass facial syndrome classification task. We tested the proposed method on a test dataset for healthy controls and a range of syndrome images for validation. We obtained the most encouraging results by adjusting the VGG-Face model. The proposed fine-tuned VGG-Face model not only demonstrated the best performance in this study, but it also performed better than other state-of-the-art pre-trained CNN models for the multi-class facial syndrome classification task. The fine-tuned model achieved both accuracy and an F1-Score of 90%, indicating significant progress in accurately detecting the specified genetic disorders.

To some extent, it can address the issue of limited data in the field of face diagnostics. In the future, we will persist in uncovering advanced deep learning models to carry out face diagnosis aided by data augmentation techniques efficiently. Additionally, more syndromes may be included in the classification problem. We anticipate that the efficiency of disease detection via facial pictures will continue to improve. Deep learning-based facial recognition shows promise in screening patients and minimizing diagnosis delays. This distinctive approach expands the current body of knowledge in the field of automated facial syndrome diagnosis.

## Figures and Tables

**Figure 1 bioengineering-11-00827-f001:**
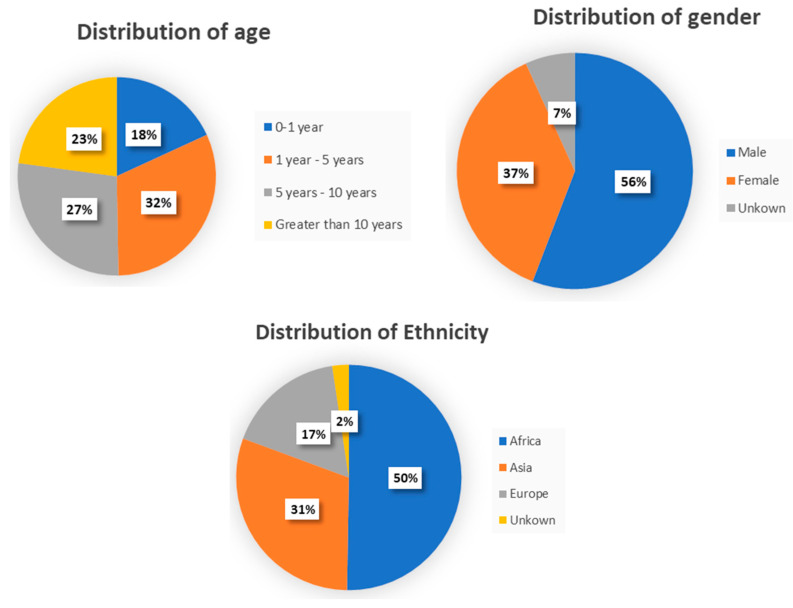
The distribution of the dataset in terms of age, gender, and ethnicity.

**Figure 2 bioengineering-11-00827-f002:**
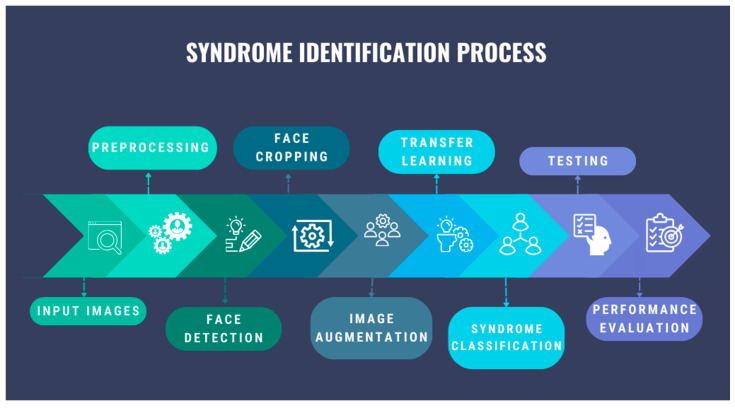
The proposed multi-class syndrome classification.

**Figure 3 bioengineering-11-00827-f003:**
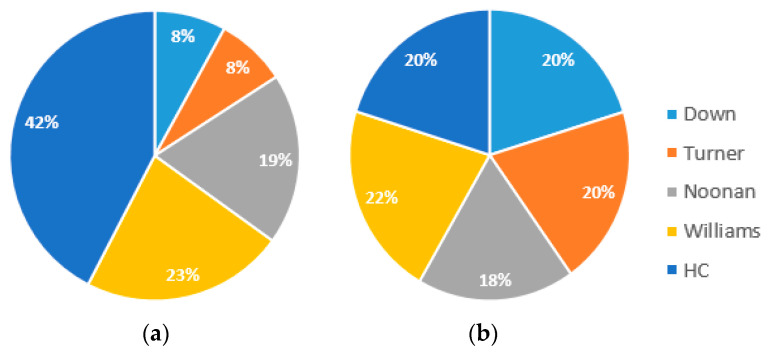
The training data distribution before (**a**) and after (**b**) data augmentation.

**Figure 4 bioengineering-11-00827-f004:**
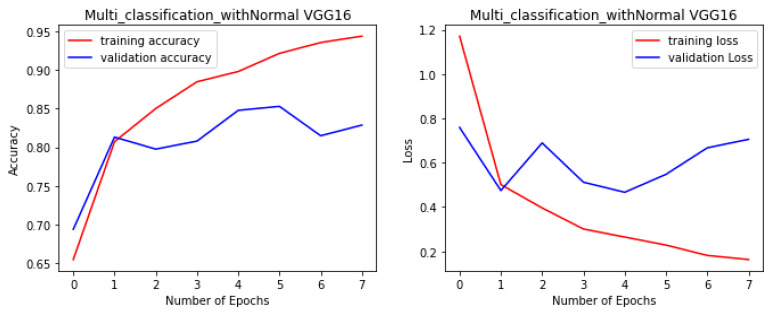
The VGG16 model performance during the training phase.

**Figure 5 bioengineering-11-00827-f005:**
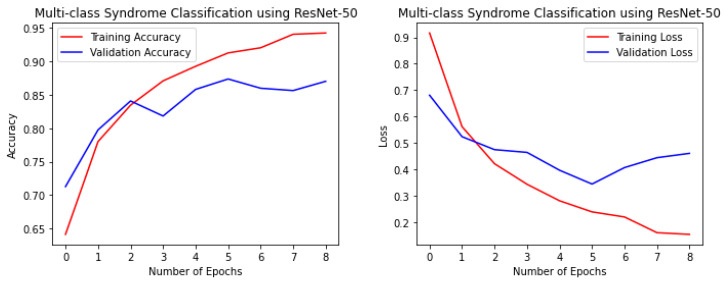
The ResNet-50 model performance during the training phase.

**Figure 6 bioengineering-11-00827-f006:**
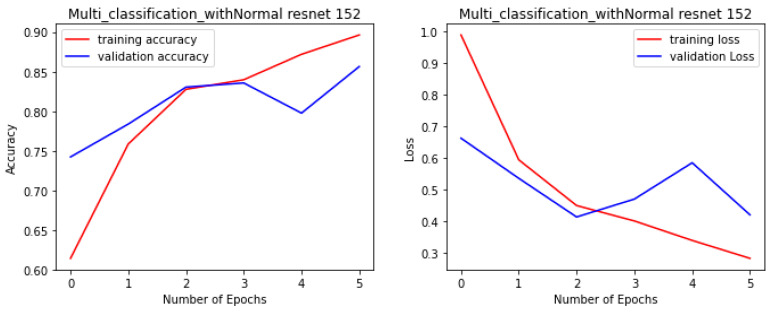
The ResNet-152 model performance during the training phase.

**Figure 7 bioengineering-11-00827-f007:**
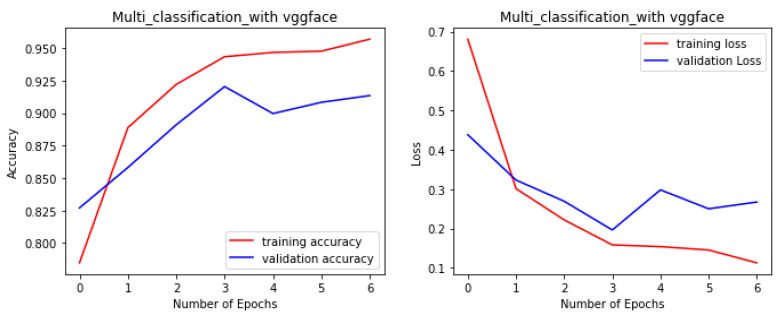
The NGG-Face model performance during the training phase.

**Figure 8 bioengineering-11-00827-f008:**
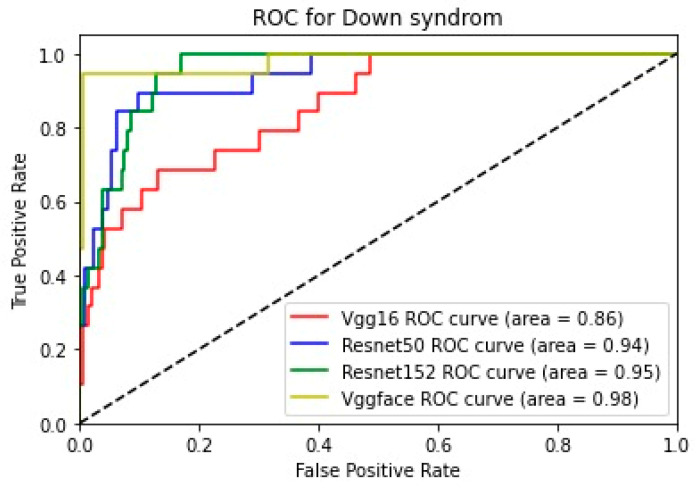
Comparative ROC curve for Down syndrome.

**Figure 9 bioengineering-11-00827-f009:**
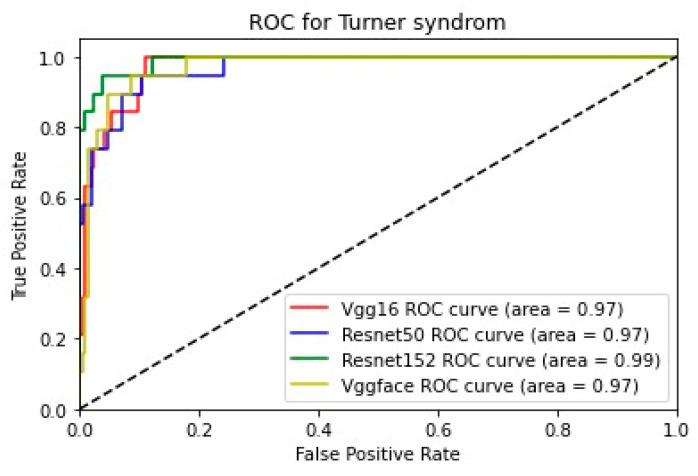
Comparative ROC curve for Turner syndrome.

**Figure 10 bioengineering-11-00827-f010:**
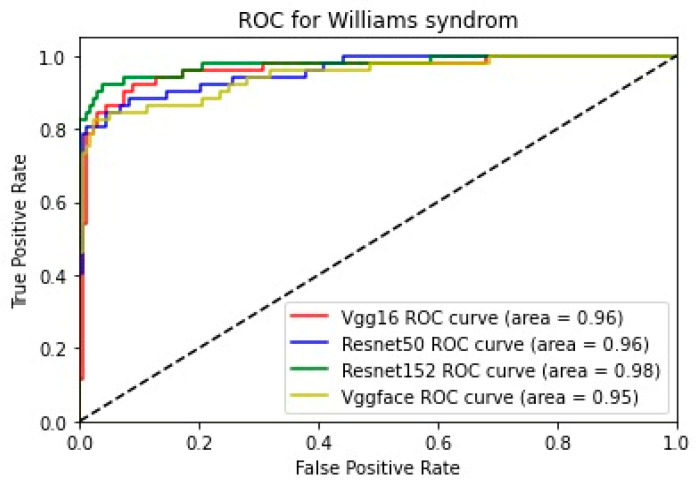
Comparative ROC curve for Williams syndrome.

**Figure 11 bioengineering-11-00827-f011:**
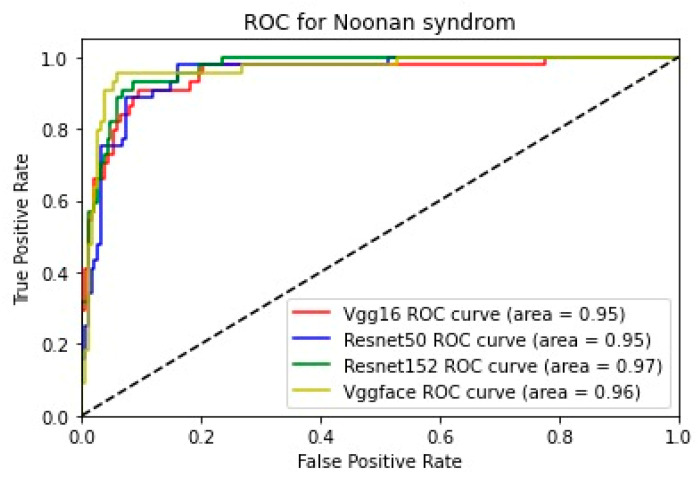
Comparative ROC curve for Noonan syndrome.

**Figure 12 bioengineering-11-00827-f012:**
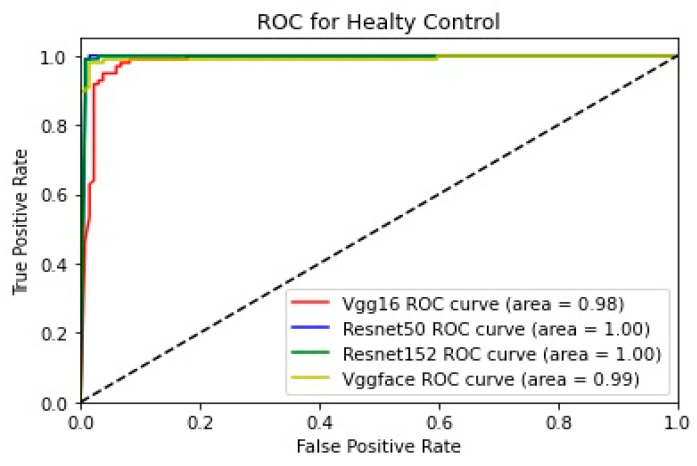
Comparative ROC curve for healthy controls.

**Table 1 bioengineering-11-00827-t001:** A summary of related automated systems for facial disease diagnosis.

Research	Dataset	Technique	Classification Problem	Results
Raza et al. (2024) [21]	1500 Down syndrome 1509 HC	VGG16	Binary	99% accuracy
Attallah (2022) [18]	70 images in each class of the following: Beta-thalassemia, Hyperthyroidism Down syndrome Leprosy HC	FaceDisNet	Binary	98.57% accuracy
			Multiclass	98% accuracy
Liu et al. (2021) [17]	104 WBS 236 HC	VGG-19	Binary	92.7% accuracy 94.0% precision 81.7% recall 87.2% F1 score 89.6% AUC
Yang et al. (2021) [14]	127 NS 163 HC 130 children with other dysmorphic syndromes.	CNN	Binary	92.01% accuracy
Pan et al. 2021 [19]	207 TS patients a 1074 HC	ResNet	Binary	95.57% AUC
Hong et al. (2021) [15]	228 children with genetic syndromes 228 HC	VGG-16	Binary	88.60% accuracy
Qin et al. (2020) [11]	148 Down syndrome 257 HC	CNN	Binary	95.87% accuracy 93.18% recall 97.40% specificity
Jin et al. (2020) [12]	70 images in each class of the following: Beta-thalassemia, Hyperthyroidism, Down syndrome Leprosy HC	VGG-Face	Binary	95% accuracy 90.9%precision 90% specificity 95.2% F1-score
			Multiclass	86.7% accuracy
Pantel et al. (2020) [10]	19 frontal face images for different of 17 syndromes 323 HC	SVM	Binary	Area under ROC curve (AUC) 0.89, 95% CI 0.87–0.92; *p* < 0.001
Liu et al. (2020) [20]	98 TS 530 HC	CNN combined with SVMs	Binary	84.95% accuracy
Kumov et al. (2020) [9]	A total of 1462 images are gathered from open sources for eight different syndromes; 204 Angelman 194 Apert 246 CDL 190 Down 158 Fragile X 142 Progeria 101 Treacher Collins 227 Williams	PCA with linear discriminant analysis (LDA) and logistic regression	Binary	The best accuracy attained was 92.5%
Gurovich et al. (2019) [13]	A dataset of over 17,000 images representing more than 200 syndromes	CNN	Binary	90.6% Top-10 accuracy
Dima et al. (2018) [8]	FERET, CAS-PEAL, LFW, AT and T for normal cases and a collection of Down faces gathered from the Internet.	PCA with SVM and KNN	Binary	The average findings were an accuracy of 98.54%, precision of 99.19%, recall of 97.6%, and specificity of 99.33%
Cerrolaza et al. (2016) [7]	A database of 145 cases, including 73 pathological patients with 15 different genetic syndromes.	Local binary pattern with LDA and SVM	Binary	SVM—RBF: AUC of 98%, accuracy of 95%, sensitivity of 93%, and specificity of 96%. SVM—linear: AUC of 98%, accuracy of 94%, sensitivity of 92%, and specificity of 96%. LDA: AUC of 97%, accuracy of 91%, sensitivity of 86%, and specificity of 96%.
Zhao et al. (2013) [6]	The dataset comprises 100 frontal facial photographs, including 50 individuals with Down syndrome and 50 healthy individuals.	SVM	Binary	The best findings were attained using SVM with radial basis function kernel, 94.6% accuracy, 93.3% precision, and 95.5%.

**Table 2 bioengineering-11-00827-t002:** A summary of the dataset.

Class Name	Total Number
Down	92
Turner	91
Noonan	219
Williams	259
Healthy controls (HC)	485
Total	1146

**Table 3 bioengineering-11-00827-t003:** Network hyperparameter optimization.

No	Parameter	Values
1	Pooling	Average
2	Batch size	32
3	Epochs	15
4	Loss-Function	Categorical
5	Early stopping	Yes
	Patience	3
6	Activation function	ReLU, SoftMax
7	Optimizer	Adam
8	Learning rate	Adaptive
	Initial_learning_rate	0.001
	Decay_steps	10,000
	Decay_rate	0.96
9	Dropout	0.2

**Table 4 bioengineering-11-00827-t004:** The performance of the VGG16 model.

Class Name	Accuracy	Precision	Recall	F1-Score	Support
Down	0.26	0.71	0.26	0.38	19
Noonan	0.70	0.79	0.70	0.75	44
Turner	0.84	0.57	0.84	0.68	19
Williams	0.87	0.80	0.87	0.83	52
HC	0.96	0.92	0.96	0.94	97

**Table 5 bioengineering-11-00827-t005:** The performance of the ResNet-50 model.

Class Name	Accuracy	Precision	Recall	F1-Score	Support
Down	0.47	0.52	0.74	0.61	19
Noonan	0.77	0.82	0.75	0.79	44
Turner	0.74	0.67	0.74	0.70	19
Williams	0.98	0.95	0.81	0.88	52
HC	1.00	0.98	1	0.99	97

**Table 6 bioengineering-11-00827-t006:** The performance of Resnet 152.

Class Name	Accuracy	Precision	Recall	F1-Score	Support
Down	0.63	0.60	0.63	0.62	19
Noonan	0.86	0.78	0.86	0.82	44
Turner	0.79	1.00	0.79	0.88	19
Williams	0.85	0.96	0.85	0.90	52
HC	1.00	0.96	1	0.98	97

**Table 7 bioengineering-11-00827-t007:** The performance of the VGG-Face model.

Class Name	Accuracy	Precision	Recall	F1-Score	Support
Down	0.95	0.75	0.95	0.84	19
Noonan	0.91	0.83	0.91	0.87	44
Turner	0.68	0.81	0.68	0.74	19
Williams	0.81	0.91	0.81	0.86	52
HC	0.98	0.98	0.98	0.98	97

**Table 8 bioengineering-11-00827-t008:** The execution performance of CNN models.

Model	Ther Ageave Training Time (Minutes)	The Average Inference Time (Seconds/Image)
VGG16	10.59	0.28
ResNet-50	58.90	0.35
ResNet-152	273.19	0.47
VGG-Face	37.36	0.22

## Data Availability

Data are contained within the article.

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
