# Peer review of "Automated Multi-Class Facial Syndrome Classification Using Transfer Learning Techniques"

_bioengineering, 2024, doi:10.3390/bioengineering11080827_

Round 1

Reviewer 1 Report

Comments and Suggestions for Authors

The paper is the application of deep learning for the early identification of genetic disorders based on facial features. The study aims to develop a more accurate and efficient method for diagnosing specific genetic disorders compared to existing techniques.

Even if the study has promising findings, addressing the following drawbacks must be offered before publication.

Major issues:

*Include information on hyperparameter tuning for each of the methods used.

*Implement cross-validation to ensure the robustness of the results across different subsets of the data.

* Include more baseline models for comparison to establish a broader context for the performance improvements. This is the only way you can prove why CNN is a better model

*Give the basic statistics of the dataset (not only the sample sizes)

*Detail the data augmentation process. Give the algorithmic details.

* Include comparative ROC-AUC charts for each dataset.

Minor issues:

*Check the references' writing format.

* Improve the quality of Figure 2.

Reviewer 2 Report

Comments and Suggestions for Authors

1. are all the photos homogeneous?any effect of angle of face, emotions or face color on the performance of the model?

2. it can be better if table 1 is written within the order of year of published?

3. why not trying deeplab, SegNet or Unet with this data?

4. for table 4-7, why not comparing against train results to see overfitting or underlearning? also it can be better to give train / test times for the models

5. fig 3 implies overfitting for some readers (or reviewers), any thought?

6. it can be better to understand if table 4-7 are based on classes not algorithms

7. table 4-7, accuracy should be added 

8. conclusion should be improved

9. finally why not comparing against other studies from the literature? is this the only usage of your dataset?

Comments on the Quality of English Language

.

Reviewer 3 Report

Comments and Suggestions for Authors

The manuscript describes a method to classify facial characteristics for genetic disorders using various neural network training models. The concept of using artificial intelligence (AI) to screen facial photographs for indications of genetic disease is good. It offers a process for expedient and low-cost medical diagnosis.

The ms has some problems. The ms is too long. The reviewer recognizes that there are many concepts and issues in the paper that require explanation. Still, if the authors were able to reduce the text, the paper would be easier to read.

 Figure 1. Word “learning” is misspelled.

In Table 2, HC = healthy controls, also written in ln 53

The ms discusses “normal cases” and normal images.

Is “normal cases” (ln 111) the same descriptor as HC (ln 53)? Please clarify. If the same, please select a single usage. The terms images, people, cases, normal, control are not well defined.

“Williams” syndrome is written multiple ways. Correct is Williams (no apostrophe s)

Section 3.4 Performance Evaluation

The formulae in the section have overworked abbreviations.

TP, FP are used for true positive, false positive, where P indicates the characteristic “positive”.

Then below for F1 score P is used for precision. Single usage for abbreviations would improve clarity.

Section 4 Results and Discussion

Ln 462 says - The goal of the study was to distinguish between normal people and those who had one of four specific genetic disorders:

Are the terms normal people, normal cases, healthy controls the same. If so better to use a single name, if not please explain the difference.

Tables 4-7. What is the column “Support”?? Why are numbers in the Support column the same for each table.

Section 5 Conclusions

Ln 594 says - We test the proposed method on both healthy individuals and a range of syndromes for validation.

What does ln 594 this mean?? Actual subjects (real people) or images from dataset?? Confusing.

Many places are confusing and difficult to understand.

The sections following Conclusions, ie author contributions, funding, data availability, are incomplete.

Overall, the paper was interesting and useful. It is a good example of using transfer learning techniques for diagnosis. The paper is too long and was difficult to follow, eg, what was accomplished. The ms needs revision for clarity. Less emphasis on the related work and more emphasis on the present work.

The language and grammar are acceptable for publication. A few typos and faulty sentence structure were present.

Recommendation. Major revision with emphasis on clarity.

Comments on the Quality of English Language

The quality of the English Language is acceptable.

There are a few places of typos and faulty sentence structure for editing. There was no problem to read the written text.

Round 2

Reviewer 1 Report

Comments and Suggestions for Authors

Thanks for the revisions which made the manuscript more readable. 

*Please give basic statistics (i.e. min, max, Mean ±Standard Deviation) of the datasets.

*Are these very good results caused by overfitting? Please discuss.

*Also, visiting the updated information for authors, I observed that your manuscript might take too long after its revision. You should find a way to reduce its length. One convenient way is to move some results to a file saved on a webpage and cite that file accordingly in your manuscript. You should focus on highlighting and including the theory in the core manuscript.

Reviewer 2 Report

Comments and Suggestions for Authors

thanks for the responses

Comments on the Quality of English Language

.

Author Response

Thank you for your valuable comments during the review rounds.

Reviewer 3 Report

Comments and Suggestions for Authors

Bioeng 3089288 Revised ms Reviewer comments

 The authors have made a good effort to revise the ms with respect to definitions.

The first version of the ms was too long. The revised is now 60 lines longer and runs 4 additional pages.

It is the editor’s decision about length. The authors added ROC plots to the paper which are useful. Maybe some of the tables and figures could be moved to supplemental materials.

 In its present revised version, it is acceptable for publication.

Author Response

Thank you for the feedback on reducing the length of the manuscript. We understand your concern about the manuscript potentially taking too long after the revision. While we appreciate the suggestion to move some results to a file saved on a webpage and cite that file accordingly, we would prefer to wait for the editor's decision or the proof-reading stage before making any major changes to the structure and content of the manuscript.

The reason for this is that the editor may have specific guidelines or preferences regarding the manuscript length and structure. Making significant changes to the manuscript at this stage, without knowing the editor's expectations, could lead to unnecessary work and potential conflict with the editor's instructions.

Moreover, during the proofreading stage, the editor and the production team may provide valuable feedback on the overall structure and length of the manuscript. This feedback could be more informed and better aligned with the journal's requirements.

Round 3

Reviewer 1 Report

Comments and Suggestions for Authors

Congratulations on the work that resulted from the revisions.